# In-Host HEV Quasispecies Evolution Shows the Limits of Mutagenic Antiviral Treatments

**DOI:** 10.3390/ijms242417185

**Published:** 2023-12-06

**Authors:** Sergi Colomer-Castell, Josep Gregori, Damir Garcia-Cehic, Mar Riveiro-Barciela, Maria Buti, Ariadna Rando-Segura, Judit Vico-Romero, Carolina Campos, Marta Ibañez-Lligoña, Caroline Melanie Adombi, Maria Francesca Cortese, David Tabernero, Juan Ignacio Esteban, Francisco Rodriguez-Frias, Josep Quer

**Affiliations:** 1Liver Diseases-Viral Hepatitis, Liver Unit, Vall d’Hebron Institut de Recerca (VHIR), Vall d’Hebron Hospital Universitari, Vall d’Hebron Barcelona Hospital Campus, Passeig Vall d’Hebron 119-129, 08035 Barcelona, Spain; sergi.colomer@vhir.org (S.C.-C.); damir.garcia@vhir.org (D.G.-C.); mar.riveiro@gmail.com (M.R.-B.); maria.buti@vallhebron.cat (M.B.); judit.vico@vhir.org (J.V.-R.); carolina.campos@vhir.org (C.C.); marta.ibanez@vhir.org (M.I.-L.); adombicaro@yahoo.fr (C.M.A.); juanignacio.esteban@vallhebron.cat (J.I.E.); 2Centro de Investigación Biomédica en Red de Enfermedades Hepáticas y Digestivas (CIBERehd), Instituto de Salud Carlos III, Av. Monforte de Lemos, 3-5, 28029 Madrid, Spain; ariadna.rando@vallhebron.cat (A.R.-S.); maria.cortese@vhir.org (M.F.C.); dtaberc@gmail.com (D.T.); frarodri@gmail.com (F.R.-F.); 3Biochemistry and Molecular Biology Department, Universitat Autònoma de Barcelona (UAB), Campus de la UAB, Plaça Cívica, 08193 Bellaterra, Spain; 4Medicine Department, Universitat Autònoma de Barcelona (UAB), Campus de la UAB, Plaça Cívica, 08193 Bellaterra, Spain; 5Microbiology Department, Vall d’Hebron Institut de Recerca (VHIR), Vall d’Hebron Hospital Universitari, Vall d’Hebron Barcelona Hospital Campus, Passeig Vall d’Hebron 119-129, 08035 Barcelona, Spain; 6Institute of Agropastoral Management, University Peleforo Gon Coulibaly, Korhogo BP 1328, Côte d’Ivoire; 7Biochemistry Department, Vall d’Hebron Institut de Recerca (VHIR), Vall d’Hebron Hospital Universitari, Vall d’Hebron Barcelona Hospital Campus, Passeig Vall d’Hebron 119-129, 08035 Barcelona, Spain

**Keywords:** quasispecies, deep sequencing, variability, rare haplotypes, fitness, mutagens

## Abstract

Here, we report the in-host hepatitis E virus (HEV) quasispecies evolution in a chronically infected patient who was treated with three different regimens of ribavirin (RBV) for nearly 6 years. Sequential plasma samples were collected at different time points and subjected to RNA extraction and deep sequencing using the MiSeq Illumina platforms. Specifically, we RT-PCR amplified a single amplicon from the core region located in the open-reading frame 2 (ORF2). At the nucleotide level (genotype), our analysis showed an increase in the number of rare haplotypes and a drastic reduction in the frequency of the master (most represented) sequence during the period when the virus was found to be insensitive to RBV treatment. Contrarily, at the amino acid level (phenotype), our study revealed conservation of the amino acids, which is represented by a high prevalence of the master sequence. Our findings suggest that using mutagenic antivirals concomitant with high viral loads can lead to the selection and proliferation of a rich set of synonymous haplotypes that express the same phenotype. This can also lead to the selection and proliferation of conservative substitutions that express fitness-enhanced phenotypes. These results have important clinical implications, as they suggest that using mutagenic agents as a monotherapy treatment regimen in the absence of sufficiently effective viral inhibitors can result in diversification and proliferation of a highly diverse quasispecies resistant to further treatment. Therefore, such approaches should be avoided whenever possible.

## 1. Introduction

HEV is a major cause of acute viral hepatitis globally, particularly in low- and middle-income countries, and its incidence is on the rise in industrialized nations. According to the WHO, approximately 20 million people are infected with HEV every year, out of which 3.3 million exhibit symptoms, and 44,000 die due to hepatic failure [1]. In Spain, the last epidemiologic study reported a 15% IgG seroprevalence in the population [2]. In addition, 1 in 3333 blood donations is positive for HEV-RNA in the Catalonia area [3]. HEV is a single-stranded positive-sense genome of 7.2 kb in length that belongs to the genus *Orthohepevirus* of the family *Hepeviridae.* HEV can be clustered genetically into eight genotypes [4]. Genotypes G1 to G4 infect humans, with G1 and G2 exclusively infecting humans through a fecal–oral route due to contamination of water supplies or food [5], whereas G3 and G4 are endemic of domestic pigs, wild boar, and deer, causing zoonotic infections in humans through consumption of uncooked or inadequately cooked processed pork meat [6,7,8] but also after transfusions of blood derivatives. In most cases, hepatitis E infecting humans causes acute self-limiting and asymptomatic infections that resolve within 2–8 weeks [9]. Occasionally, a serious disease known as fulminant hepatitis (acute liver failure) develops, which can be fatal [10]. In immunocompromised patients, such as solid organ transplant recipients, those receiving immunosuppressors to prevent organ rejection, patients infected by the human immunodeficiency virus (HIV) with a low CD4+ cell count, and patients receiving chemotherapy for hematological disorders G3 and G4 infection, can have a persistent infection named chronic hepatitis E [11], which is diagnosed after 3 months of continuous viremia [12]. 

Current clinical guidelines recommend an initial reduction in immunosuppression as the first step to manage chronic HEV, especially in solid organ-transplanted patients [9]. Nevertheless, if HEV is not eliminated, a 3-month ribavirin (RBV) monotherapy with a weight-adjusted dose or a dose based on glomerular filtration rate is the standard treatment. The sustainable virological response (SVR) has been estimated up to 78% [13,14]. When HEV relapse or the first treatment fails, a re-treatment with ribavirin for 6 months has to be carried out. However, there is no knowledge about the mechanism of action of RBV in HEV clearance, the principal hypothesis being the depletion of cellular GTP pools [15] or the lethal mutagenesis [16,17,18]. On the other hand, PEG-Interferon-α has shown effectiveness in HEV clearance in liver-transplanted patients, but it is contraindicated in other transplants due to its immune activation mode of action [9]. Therapeutic alternatives have been proposed, such as the combination of ribavirin with sofosbuvir, but negative results have been reported. The lack of consensus about RBV doses and treatment times, together with the apparition of mutations associated with RBV resistance, evidences the need for new antiviral treatments [19,20]. 

NGS has been reported as the most accurate methodology to study highly variable viruses such as HEV [16]. HEV is a is a highly variable virus, showing a rate of fixation of mutations of 1.41–1.72 × 10^−3^ substitutions/nucleotide/year [21], causing continuous production of variants during infection and generating a complex mixture of different but closely related genomes known as quasispecies [22,23]. During chronic infection, typically, a dominant strain (wild type) is detectable within the viral quasispecies along with strains that are present at lower frequencies [24]. We have recently showed that a quasispecies partition into fitness fractions (QFF, quasispecies fitness fractions) provides a valuable visualization, with biological/clinical implications, that contributes to explain the molecular changes in the composition of a quasispecies over time [16]. 

Here, we report the deep-sequencing study of samples sequentially collected from a HEV chronically infected patient that received three treatments with RBV. We found profound discrepancies between the nucleotide population patterns (genotype level) compared to the protein patterns (phenotype level) that have important implications in the evolution of the viral infection, disease progression, and treatment strategies, having general implications for any antiviral treatment prescription. 

## 2. Results

In this study, we observe the cumulative effect of three different ribavirin treatment regimens, with EOTs and large periods with no treatments in between. The timeline with interventions, samplings, and viral loads is shown in Table 1 and Appendix A. The evolution in the quasispecies is studied at the genotype and phenotype levels with the following tools: (a) UPGMA tree of master sequences (Figure 1); (b) UPGMA tree of quasispecies (Appendix A); (c) quasispecies structure by the method of the fitness fractions (Figure 2) and quasispecies diversity by Hill number profiles (Appendix A); (d) evolution in the fraction of synonymous reads and haplotypes to the master phenotype (Figure 3); (e) evolution of the substitution and mutation loads in the quasispecies, corresponding to synonymous haplotypes to the master phenotype (Figure 4); (f) distribution of the top 10 haplotypes and phenotypes in each sample (Figure 5); and (g) characterization of the impact of emerging phenotypes in protein functionality (Appendix A). Additional analyses have been carried out and are shown in the Appendix A.

### Traits in Quasispecies Evolution

The first treatment lasted from day 447 to 536, with samples at 448 (S03), 450 (S04), and 456d (S05). Viral load dropped slightly from 4.00 × 10^6^ to 2.78 × 10^6^ and 1.17 × 10^6^. The master haplotype decreased in frequency (34.58%, 31.49%, and 24.72%). The quasispecies showed steady fraction <0.1% in QFF, increasing in 0.1–1% and emergent (>1%). Similar patterns emerged at the phenotypic level (Figure 2). RBV’s effect could be seen in declining similarity between 450d/448d and 456d/450d (Appendix A) and the increase in intra-quasispecies diversity (Appendix A).

After this EOT and during the absence of RBV treatment, the same master haplotype and phenotype were maintained between samples 1087d (S06) and 1162d (S07), showing similar frequencies (32.47%, 26,81% and 70.30%, 64.77%) as before (Figure 1). The quasispecies structure and diversity remained similar in terms of QFF (Figure 2) and inter-sample similarity and distance (Appendix A). Amino acid diversity increased, whereas nucleotidic diversity remained steady (Appendix A), and the fraction of synonymous reads decreased (Figure 3).

On day 1162 (S07), the patient entered a clinical trial with RBV 200/400 mg daily. No samples were taken in this 20-week period. After the second EOT (no RNA-VHE negativization), at day 1358 (S08), the master haplotype and phenotype frequencies increased significantly (59.67% and 75.05%, Figure 1). Quasispecies diverge from the previous sample (1162 (S07), previous to the trial) both genetically and phenotypically (Appendix A). The QFF showed a tiny fraction of emergent haplotypes and reduced rare haplotypes (Figure 2). The fraction of synonymous reads expressing the master phenotype increased significantly, from 64.7% to 75.0%, whereas the fraction of synonymous haplotypes decreased from 41.6% to 30.4% (Figure 3), suggesting that a new quasispecies might have appeared as a consequence of the proliferation of new fitness-enhanced haplotypes and phenotypes caused by the mutagenic treatment. 

After 56 days of no treatment, in 1414d (S09), a different master haplotype and phenotype emerged (6.1% and 62.7%). Genetic QFF showed proliferation of haplotypes, especially emergent ones and the 0.1 fitness fraction, whereas only emerging phenotypes increased at this time point (Figure 2). Interestingly, the similarity with the previous sample was low for haplotypes but relatively high for phenotypes (functional level). Both nucleotide and amino acid diversity (Appendix A) increased substantially due to the proliferation of new genomes progressing from very low frequencies.

Finally, after 80 weeks, a new treatment of 400 mg daily for 24 weeks is started. Three samples at 2065 (S13), 2096 (S15), and 2135 days (S18) showed different master haplotypes at low frequencies (4.58%, 4.24%, and 4.12%) but the same phenotype (40.96%, 35.42%, and 66.86%) (Figure 1). Quasispecies clustered with the 1414d (S09) sample but showed differences (Appendix A). Genetic QFF resembled the 1414d structure, but phenotypic QFF indicates emerging phenotypes at the expense of the master, except for the last sample (Figure 2). The similarity in sequential sample haplotypes remained low, whereas phenotypes showed values around 0.5 (Appendix A). Viral loads stayed above or slightly below 5 logs, indicating a poor treatment response.

In summary, the QFF at the genetic level shows a complex quasispecies, with multiple haplotypes able to compete with the master, and eventually replace it, with a steady and very important fraction of reads for haplotypes <1% in frequency, which remain at the same level despite the time between them. The QFF at the phenotypic level shows also an important fraction of emerging phenotypes, which represent functional alternatives to the master. To further clarify this situation and the mutagenic effects of the treatment, the reads of haplotypes synonymous to the master phenotype in each sample were aggregated according to the number of substitutions in the haplotype with respect to the master haplotype in each sample. The result is shown in Figure 4. All samples until 1162d (S07) show a high fraction of single mutant haplotypes (m1), similar to the frequency of the master haplotype (m0), and a growing fraction of double mutant haplotypes (m2). At 1358d (S08), the master and the single mutants are the only important fractions, but tiny fractions of higher order mutants are observed. These higher order mutants were undoubtedly generated during the treatment in the clinical trial and could proliferate from very low frequencies to appear visible in the 8 weeks from EOT. The 1414d sample (S09), 8 weeks later, confirmed this proliferation with sensible fractions at multiplicities 1 to 5 (Figure 4). The last three samples (2065d, 2096d, and 2135d), again under treatment but with relatively high viral loads, showed the further proliferation of old mutants and newly generated with multiplicities up to 15 substitutions, observed in this ORF2 amplicon of 363 bp.

Regarding the pattern of substitutions with respect to the consensus one, we observed that the majority of substitutions were transitions. Interestingly, substitutions such as C → T/C and G → A/G, associated with the ribavirin effect, were notably prevalent in the last four samples (Appendix A). This pattern is accentuated when specifically analyzing synonymous nucleotide mutations (Figure 6).

Beyond the synonymous haplotypes, of whatever multiplicity, generated during the treatment, the conservative substitutions expressing alternative fitness-enhanced phenotypes should be considered. The distribution of the 10 top haplotypes and phenotypes in each sample are represented, in descending order of frequency, in Figure 5. The last four samples show the top ten haplotypes at low and very similar frequencies, indicating similar fitness. The corresponding phenotype distributions show a prominent master but a series of alternative phenotypes at sensible frequencies. Analyzing the amino acid changes that these haplotypes introduce with respect to the master phenotype in the d0 sample, we see that all could be the product of a single nucleotide substitution, and the resulting amino acids have similar chemical characteristics to the wild type (Table 2 and Appendix A) compatible with moderate-to-fitness-enhanced alternative phenotypes.

## 3. Discussion

In our study, we observed the cumulative effect of three different ribavirin treatment regimens (currently, the only clinical option), with EOTs, and large periods with no treatment in between, and we compared the effect at the nucleotide (genotype) and amino acid (phenotype) level. The master haplotype at mid frequencies (24.7–44.3%) was maintained up to 1162d; a new master haplotype appeared at 1358d with higher frequency (59.7%); and a new master haplotype was generated in each successive sample, with very low frequencies (4.1–6.1%), showing an unstructured collection of genomes when the viral population showed insensitivity to increase in RBV amounts (1414d to 2135d) at the nucleotide level. Interestingly, first treatment at the naïve-treatment time, with 200 mg RBV, seemed to be effective since RNA achieved negativization 90 days after starting treatment, at least with the diagnostic sensitivity used at that time using plasma sample, suggesting that lethality could have been achieved. However, by stopping treatment, possibly too early in time, the virus had not reached complete extinction. Basal viral replication likely generated a drifting mutant cloud with continuous loss of the master sequence due to the advantage of genomes lying on a high fitness plateau [25,26] (reviewed in Tejero et al. [27]) and helped the population to move through the sequence space changing the master sequence in 1358d (frequency of 59.7%) and a subsequent jump to a new master sequence. The most breaking result in our study is that, at the nucleotide level, dynamics of viral quasispecies look like there is a loss in the fitness of the master sequence, but, at the phenotype level, once the virus found the new master sequence at day 1414d, it becomes highly dominant (66.86% at day 2135d) despite the increase in the amount of RBV provided. Quasispecies theory suggests that viruses could also achieve resistance by moving to flatter regions of the fitness landscape, where the density of neutral mutations is higher (survival of the flattest) [28,29].

Treating chronic HEV infection, primarily affecting immunocompromised patients, is challenging due to the absence of direct-acting antiviral drugs for HEV. The virus’s high mutation rate, and high viral loads in these patients pose an added problem, reducing the effectiveness of repurposed drugs like sofosbuvir for antiviral treatment. Currently, Ribavirin (RBV) is the only effective drug against HEV due to its multiple mechanisms of action [30], including a putative lethal mutagenesis effect [17,31]. 

Lethal mutagenesis is based on the idea that RNA viruses replicate at an exceptional high mutation rate, nearing what’s known as error catastrophe [32,33]. Eigen’s original theory proposes that a quasispecies can remain stable despite a high mutation rate. However, even small increases in mutation rate can disrupt this equilibrium, leading to loss of the master sequence and meaningful genetic information due to a surge in errors [34].

Initially, error catastrophe described the deterioration of cellular functions in the context of aging [35]. It was later adapted to the quasispecies theory [36], signifying a critical transition from a structured distribution of viral genomes, usually dominated by a master sequence, to an unstructured collection of genomes, without a master sequence. Lethal mutagenesis aims to render a viral population nonfunctionally by elevating the average mutation rate beyond a specific threshold, often achieved through the use of mutagenic agents. 

RBV-mediated lethal mutagenesis (extinction through an increase in mutant spectrum complexity and decrease in specific infectivity) has been reported for HCV in cell cultures [37] and in other viruses [17]. Early studies on foot-and-mouth disease virus (FMDV) revealed that viral populations with high fitness and large population sizes exhibited reduced susceptibility to mutagenic agents, leading to delayed virus extinction [38]. In our study, the sustained viral loads along the treatments, with low effectivity, could be, in part, explained by the presence of a highly fit viral population.

Previous studies have demonstrated that achieving extinction in high-fitness viral populations requires a strategic combination of mutagenic agents and nonmutagenic inhibitors [39,40]. This approach was found necessary against the resilient nature of such quasispecies. Actually, a higher suppressive effect of a sequential administration of an antiviral non-mutagenic inhibitor, followed by a mutagenic agent than the converse (first a mutagenic agent and then an inhibitor) or the corresponding combination (inhibitor and mutagen-administered together), has been documented [41,42]. The rationale is that the administration of the inhibitor will produce a decrease in viral load, which will render the system more susceptible to mutagenesis-mediated extinction, allowing expression of interfering activities associated with the mutagenized spectrum of mutants [43].

The results in this study are consistent with the development of resistance to the treatment. Whether specifically to RBV, the result of a highly fit and complex quasispecies, or both at the same time, is unknown. The presence of variants over the full HEV polymerase, with these samples, is under study. On the other hand, viral fitness has been described as a key factor in resistance to treatments in HCV, both mutagenic and direct acting agents [16,44]. 

A similar case of an HEV patient treated with different regimens of RBV has been recently described [16] with the same consequences: high final viral loads of a non-responding quasispecies, highly diverse in genetic and phenotypic terms. Furthermore, these results could be theoretically anticipated. Only two factors are required to achieve this final situation: (a) an enhanced production of mutants (mutagenic treatment, like RBV) and (b) a replicating system (quasispecies in a host) at a pace (viral load) able to keep selecting the most-fit variants while removing the weakest. On the other hand, each treatment discontinuation with RNA+ will bring to the proliferation of the new fitted variants produced during the treatment, resulting in a highly diverse quasispecies resilient to new treatments. This study shows a realization of this scenario: quasispecies dynamics in pure state.

## 4. Materials and Methods

### 4.1. Patient Data

Sequential plasma samples were collected at different time points (Table 1, Appendix A) from a 62-year-old patient chronically infected by HEV. His medical records included lung transplantation for interstitial lung disease and sequential kidney transplantation due to end-stage chronic kidney disease. Diagnosis of chronic hepatitis E was established based on detectable HEV RNA after long-term transaminases increased. The first RBV treatment was adjusted to kidney function to 200 mg/day for three months (12 weeks), resulting in an undetectable HEV RNA. One year later, there was a relapse, leading to a second RBV treatment of alternating 200 and 200/400 mg/day for 6 months (24 weeks), with a posterior 12 months (48 weeks) monitorization. The second treatment was not capable of eliminating the HEV but achieved hepatic biochemistry normalization and platelet count elevation. Finally, a third treatment of 400 mg/day for 6 months was applied due to hepatic decompensation and ascites appearance. This third regime was able to reduce ascites but could not eliminate HEV infection nor SVR, keeping a sustained viremia.

### 4.2. RNA Extraction and Amplification

RNA was extracted using the QIAamp RNA Viral MiniKit (QIAGEN, Hilden, Germany) following the provider protocol. RNA was retrotranscribed and subsequently amplified using SuperScript™ III One-Step RT-PCR System with Platinum™ Taq High Fidelity DNA Polymerase (Termo Fisher Scientific, Carlsbad, CA, USA), and two external ORF2 primers: FW 5′CCGACAGAATTGATTTCGTCGGC3′ and RV 5′ACTCCCGRGTYTTACCYACCTT3′. RT-PCR was performed at 50 °C for 30 min, followed by 7 min a 94 °C for RT enzyme inactivation, and successive cycling of 94 °C 10 s denaturation, 54 °C 30 s annealing, and 68 °C 1 min 30 s elongation, with a final elongation of 7 min. Nested PCR was carried out with FastStart High Fidelity PCR System, dNTPack (Roche, Basel, Switzerland), according to manufacturer indications using the internal primers FW 5′ GTCGTCTCAGCCAATGGCGAGCC3′ and RV 5′CASARAANGTCTTNGARTACTGCT3′, with annealing and elongation temperature at 50 °C and 72 °C, respectively (Figure 7). PCR product was purified using KAPA Pure Beads (Kapa Biosystems, Roche, Pleasanton, CA, USA). Quantification was carried out by Qubit fluorometry using a double-stranded DNA High Sensitivity Assay Kit (Termo Fisher Scientific, Carlsbad, CA, USA).

### 4.3. Library Preparation and Illumina Sequencing

Library preparation was carried out using KAPA HyperPrep Kit (Kapa Biosystems, Roche, Pleasanton, CA, USA), according to manufacturer instructions. Samples were single indexed using the SeqCap Adapter Kit A/B (Nimblegen, Roche, Pleasanton, CA, USA), and DNA purification was performed through KAPA Pure Beads (Kapa Biosystems, Roche, Pleasanton, CA, USA). Final library concentration was assessed by qPCR using the KAPA Library Quantification Kit (KapaBiosystems, Roche, Pleasanton, CA, USA). Library was loaded onto Miseq Illumina platform through MiSeq Reagent Kit v3 600 cycles (Illumina, San Diego, CA, USA).

### 4.4. Processing the Sequencing Data

The aim of the sequencing data treatment is to discard error-bearing reads while preserving full-length read integrity so that haplotypes that completely cover the amplicon with their respective frequencies were incorporated. The steps in this process have been previously described [16,45] and may be summarized as follows:Obtain Fastq files with Illumina 2 × 300-bp paired-end reads;Recover full amplicon reads with FLASH [46] (minimum 20-bp overlap, maximum 10% mismatches). The 300-bp reads, when overlapped, result in reads covering complete ~400–500 bp amplicons;Remove full reads with 5% or more bases below a Phred score of Q30;Demultiplex and trim primers (max three differences accepted);Collapse reads (molecules) to haplotypes (amplicon-genomes) and their frequencies (read counts);Multiple alignment of all haplotypes in each sample/amplicon;Remove all haplotypes that are not common to both DNA strands and supported at least by 5 reads;Remove insertions in master haplotypes and repair single gaps in remaining haplotypes. Recollapse to haplotypes and vectors of frequencies;The haplotypes in each sample/amplicon are translated to phenotypes and recollapsed to obtain the set of phenotypes with corresponding frequencies (read counts).

### 4.5. Bioinformatic Procedures

The haplotypes/phenotypes and corresponding frequencies resulting from the sequencing data are the basis of subsequent computations. A detailed description of these methods is provided in the Appendix A. Briefly, the computations took into account sample size dependence of diversity indices by repeated resampling to the reference coverage (147,000 reads). The quasispecies structure and diversity was studied by the method of the Quasispecies Fitness Fractions and the Hill Numbers Profile. Distances between quasispecies were computed by the Matoshi-Nei method, using raw genetic distances between pairs of haplotypes at the genetic level, or the Grishin distances based on BLOSUM80 matrix values at the phenotypic level. 

### 4.6. Software and Statistics

All computations were performed in R (v4.2.2) [47] with in-house scripts, using the Biostrings [48], ShortRead [49], and QSutils [50] packages from Bioconductor [51], as well as ape [52], bios2mds [53], tidyverse [54], and ggplot2 [55].

## 5. Conclusions

The treatment with mutagenic agents concomitant with high viral loads resulted in an accelerated evolution of the quasispecies with the selection of fitness-enhanced haplotypes and phenotypes, creating fitness-enhanced and extremely diverse quasispecies that are resistant to further treatments. The prescription of mutagens should only be advised when combined with efficient viral inhibitors, in order to keep the replication as low as possible and to avoid the selection of better-fit variants. Even in this case, it should be prescribed only to patients able to follow the treatment until RNA negativization, with no adverse effects that could recommend an early discontinuation of the treatment. The risk is to end up with a better adapted quasispecies, resilient to further treatments.

## Figures and Tables

**Figure 1 ijms-24-17185-f001:**
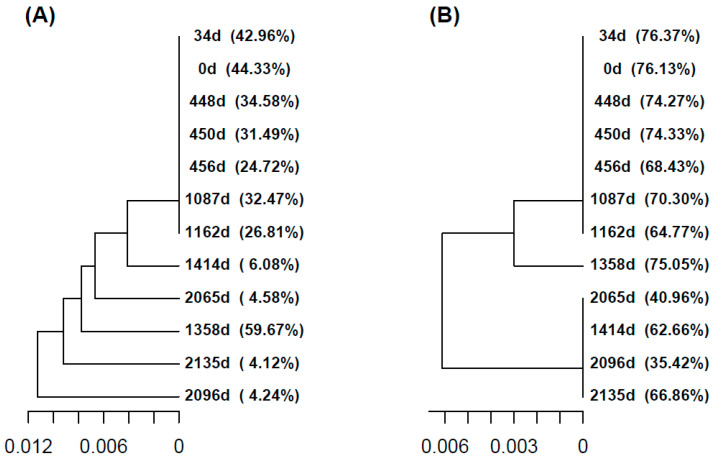
(**A**) UPGMA tree with the master haplotypes of all samples, at the nucleotide level. (**B**) Corresponding tree with the master phenotypes (amino acid level) of all samples in the study. Samples are labelled as days since first evidence. The observed frequencies of each sequence are expressed as percentages.

**Figure 2 ijms-24-17185-f002:**
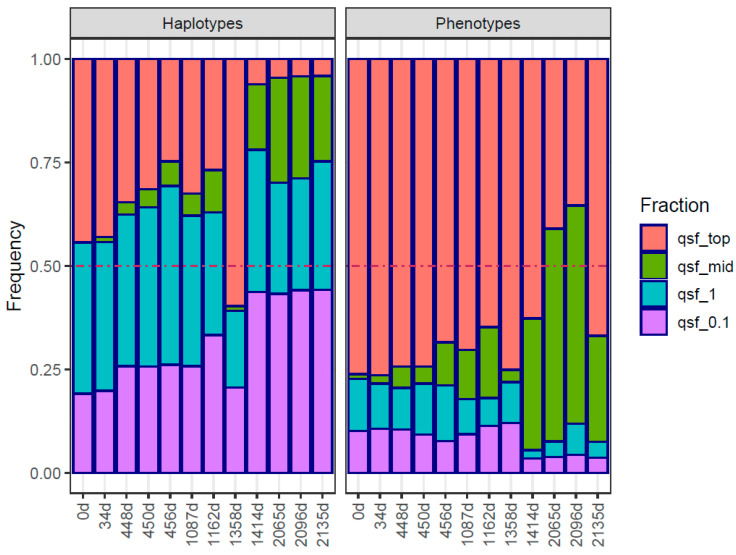
Quasispecies Fitness Fractions for each sample, at the genetic level on the left, and at the phenotypic level on the right. qsf_top: fraction of reads belonging to the master; qsf_mid: fraction of reads belonging to emerging haplotypes; qsf_1: fraction of reads of rare haplotypes; qsf_0.1: fraction of reads of very rare and defective haplotypes. The dash-dot line at 0.5 helps in visualizing the masters of either type with a frequency above 50%.

**Figure 3 ijms-24-17185-f003:**
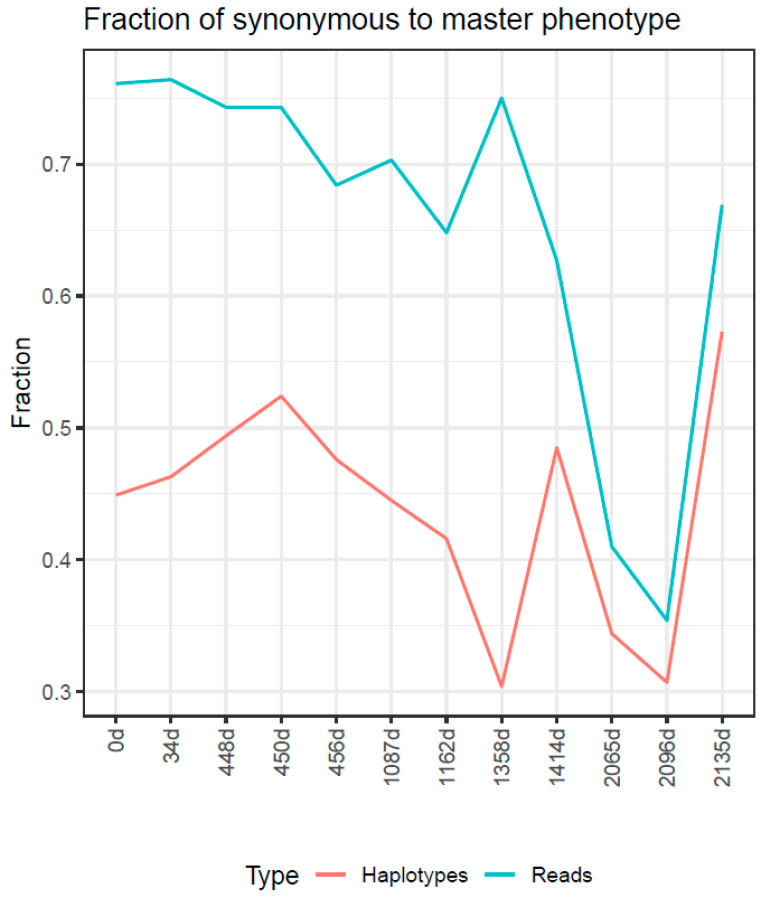
The fraction of synonymous reads to the master phenotype of each sample (turquoise) and the fraction of synonymous haplotypes to the master phenotype of each sample (orange).

**Figure 4 ijms-24-17185-f004:**
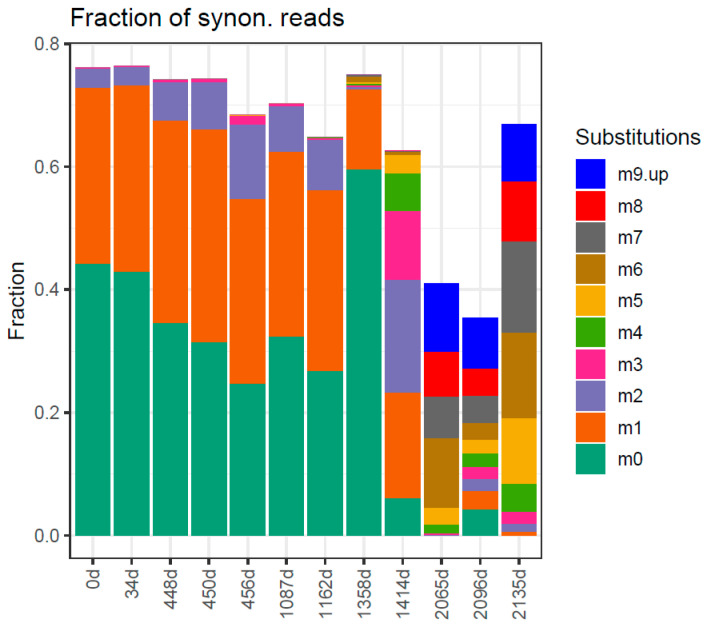
Aggregation of reads of synonymous haplotypes to the master phenotype, according to the number of substitutions with respect to the master haplotype in each sample. m0: fraction of reads for the master; m1: fraction of reads with one substitution with respect to the master haplotype; and so on.

**Figure 5 ijms-24-17185-f005:**
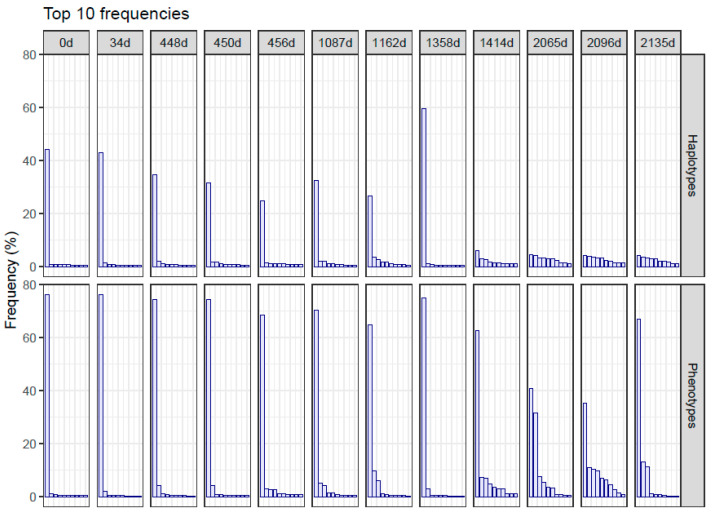
Frequencies of the top 10 haplotypes and phenotypes, shown in decreasing order.

**Figure 6 ijms-24-17185-f006:**
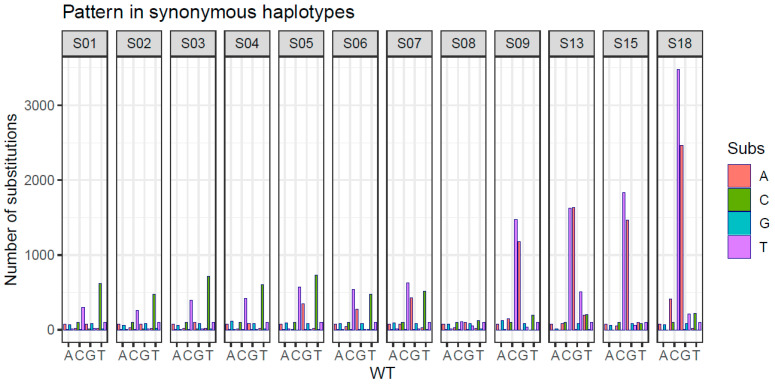
Substitution pattern in synonymous haplotypes to the wild type (consensus synonymous haplotype in each sample).

**Figure 7 ijms-24-17185-f007:**
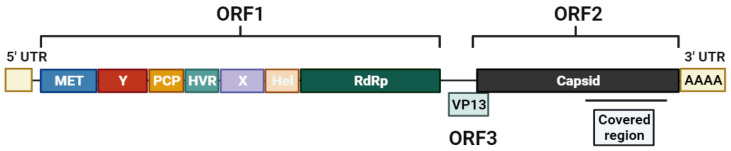
HEV genomic structure scheme pointing out the covered region. Created with BioRender.com.

**Table 1 ijms-24-17185-t001:** Treatment regimens and samplings during patient disease. Periods with treatment are shown with a light-grey background. RBV: Ribavirin; EOT: End of Treatment.

Intervention *	ID ^a^	Date	Days ^b^	Weeks ^c^	CumDays ^d^	Viral Load
Infection		October 2012				
1st evidence	S01	13 November 2012	0	0.0	0	5.04 × 10^6^
	S02	15 December 2012	34	4.9	34	6.84 × 10^6^
RBV 200 mg		3 February 2014	413	59.0	447	
	S03	4 February 2014	1	0.1	448	4.00 × 10^6^
	S04	6 February 2014	2	0.3	450	2.78 × 10^6^
	S05	12 February 2014	6	0.9	456	1.17 × 10^6^
EOT		3 May 2014	80	11.4	536	RNA−
Relapse		1 January 2015	243	34.7	779	RNA+
	S06	5 November 2015	308	44.0	1087	3.56 × 10^6^
RBV 200/400 mg	S07	19 January 2016	75	10.7	1162	1.30 × 10^7^
EOT		5 July 2016	168	19.7	1300	RNA+
	S08	2 August 2016	28	8.3	1358	2.00 × 10^4^
	S09	27 September 2016	56	8.0	1414	6.93 × 10^5^
RBV 400 mg		9 April 2018	559	79.9	1973	1.27 × 10^7^
	S13	10 July 2018	92	13.1	2065	1.51 × 10^5^
	S15	10 August 2018	31	4.4	2096	8.22 × 10^4^
	S18	18 September 2018	39	5.6	2135	7.50 × 10^4^
EOT		27 November 2018	70	10.0	2205	8.20 × 10^4^
		2 March 2019	95	13.6	2300	7.00 × 10^6^

* Intervention includes the different treatments, samplings, and viral load monitorizing in which sample was taken. ^a^ Sample ID. ^b^ Days since previous row. ^c^ Weeks since previous row. ^d^ Days since first evidence.

**Table 2 ijms-24-17185-t002:** Variants observed in emerging haplotypes of the last four samples with respect to the d0 phenotype. WT: wild type amino acid (in d0 phenotype). Var: observed mutation. N: number of phenotypes in which this mutation has been observed, this includes any master or emerging phenotypes of any of the four samples. Fitch: distance of Fitch as the minimum number of nucleotide substitutions required for the mutation. Grantham: chemical distance between the two amino acids. qGrantham: quantile of the corresponding Grantham distance.

WT	Var	N	Fitch	Grantham	qGrantham
T	N	27	1	65	0.2579
T	A	20	1	58	0.2105
G	S	3	1	56	0.2000
V	I	3	1	29	0.0789
A	T	2	1	58	0.2105
E	K	2	1	56	0.2000
I	M	2	1	10	0.0105
L	F	2	1	22	0.0368
A	V	1	1	64	0.2421
F	Y	1	1	22	0.0368
I	V	1	1	29	0.0789
S	A	1	1	99	0.5368
S	P	1	1	74	0.2947
V	A	1	1	64	0.2421
Y	H	1	1	83	0.3421

## Data Availability

The genomic nucleotide sequences included in this study are being submitted in the GENBank repository database as Bioproject ID PRJNA1038697.

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
