# Peer review of "In-Host HEV Quasispecies Evolution Shows the Limits of Mutagenic Antiviral Treatments"

_ijms, 2023, doi:10.3390/ijms242417185_

Round 1
Reviewer 1 Report
Comments and Suggestions for Authors
Author Response
1- I would really like a schematic of the virus with the regions that are sequenced highlighted. It would be nice to see how it lies within the genome and the where the genes are located.
Answer: Thank you for the recommendation. We have inserted Figure 1 in the Methods, showing the genes and the studied region.
2- Could the authors comment on the nucleotide mutations that arise that do not change the amino acid phenotype. Is there a basis for a specific nucleotide or nucleotides? For example say a G to C change or whatever?
Answer: Thank you for your insightful comment. Actually, we added the global pattern of substitutions in relationship to the consensus within each quasiespecies in the Supplementary Table 6 and elaborated in the Supplementary Results. Our study revealed a noteworthy prevalence of C→T / C and G→ A / G mutations, specifically associated with RBV, particularly in the last four samples. Following your clever recommendation, we have conducted an additional analysis focusing exclusively on synonymous mutations, confirming the data previously shown.
In response to your feedback, we have included an additional paragraph and a corresponding figure in Results lines 308-312. This new sentence emphasizes, “Regarding the pattern of substitutions with respect to the consensus one, we observed that the majority of substitutions were transitions. Interestingly, substitutions such as C→T / C and G→ A / G, associated with the ribavirin effect, were notably prevalent in the last four samples (Suppl. Table 6). This pattern is accentuated when speficically analysing synonymous nucleotide mutations (Figure 7).”
This addition not only reinforces our initial observations but also provides a more comprehensive understanding of our study.
Reviewer 2 Report
Comments and Suggestions for Authors
Colomer-Castell et al. aimed to report the in-host HEV quasispecies evolution in a chronically 24 infected patient who was treated with three different regimens of RBV. This is a very interesting study and I think it should be shared with the international community. I would just ask to authors to correct some papers’ elements.
1. Introduction. Please, report your national epidemiological data if available. If not, add a specific comment on that.
2. Methods. Methods should be reported after introduction and before your results. Please, revise accordingly.
3. Table 1. The title is too long. Please, revise as: ‘treatments, samplings, and viral loads in patients included in our study. ‘ID: 110 sample ID, Days: days since previous row, sample or intervention; Weeks: weeks since 111 previous row; CumDays: days since first evidence’: this is a legend, should be putted after the table. Please, specific also your abbreviations (e.g. EOT: end of treatment, RBV: ribavirine, and so on)
4. Informed consent statement is reported twice, please correct.
5. References. References are not all in line with the Publisher guidelines. Please, revise accordingly (e.g. all authors should be reported in the reference; in line 235 there is also a typo with a reference reported in apex).
6. Discussion. Before commenting chronic HEV challenging in treatment and so on, please comment your results as per Literature standards.
7. Line 299. ‘59,7%’ should be ‘59.7%’
8. From line 324 to line 333. Separate this part using it as ‘Conclusion’.
Comments on the Quality of English LanguagePlease, use shorter periods when writing, or readability will be reduced.
Author Response
- Please, report your national epidemiological data if available. If not, add a specific comment on that.
Answer: Indeed, the introduction was lacking epidemiological data, thank you for the comment. To address this gap, we have added the sentences: “In Spain, the last epidemiologic study reports a 15% IgG seroprevalence in the population[2]. In addition, 1 in 3333 blood donations is positive for HEV-RNA[3].” in lines 47 and 48 using the data reported by the Spanish Health Ministry and Catalonia Blood and Tissue Bank”.
- Methods. Methods should be reported after introduction and before your results. Please, revise accordingly.
Answer: Thank you for this comment. It has been revised and reordered.
- Table 1. The title is too long. Please, revise as: ‘treatments, samplings, and viral loads in patients included in our study. ‘ID: 110 sample ID, Days: days since previous row, sample or intervention; Weeks: weeks since 111 previous row; CumDays: days since first evidence’: this is a legend, should be putted after the table. Please, specific also your abbreviations (e.g. EOT: end of treatment, RBV: ribavirine, and so on)
Answer: We thank rev#2 for pointing out. We have simplified the title to “Treatment regimens and samplings during patient disease. Periods with treatment are shown with light grey background. RBV: Ribavirin; EOT: End of Treatment.” including the abbreviations in lines 190-191. All the explanations in the header have been put after the table using superscript (see lines 214-219).
- Informed consent statement is reported twice, please correct.
Answer: Corrected (see lines 471-472)
- References. References are not all in line with the Publisher guidelines. Please, revise accordingly (e.g. all authors should be reported in the reference; in line 235 there is also a typo with a reference reported in apex).
Answer: Thank you for the comment. References have been included using Mendeley citation assistant using the IJMS citation style. Do you mind if do we ask the editorial board whether we have to change the style?
- Discussion. Before commenting chronic HEV challenging in treatment and so on, please comment your results as per Literature standards.
Answer: Corrected. We have commented our results in the first place (lines 357-380), and after we start discussing about the chronic HEV challenge.
- Line 299. ‘59,7%’ should be ‘59.7%’
Answer: Corrected (see line 373)
- From line 324 to line 333. Separate this part using it as ‘Conclusion’.
Answer: Corrected (see lines 436-446)